# Towards Holistic System Models Including Domain-Specific Simulation Models Based on SysML

**Yizhe Zhang \*****, Gregor Hoepfner** **, Joerg Berroth** **, Gerwin Pasch**  **and Georg Jacobs** 

Institute for Machine Elements and Systems Engineering, RWTH Aachen University, 52062 Aachen, Germany;
gregor.hoepfner@imse.rwth-aachen.de (G.H.); joerg.berroth@imse.rwth-aachen.de (J.B.);
gerwin.pasch@imse.rwth-aachen.de (G.P.); georg.jacobs@imse.rwth-aachen.de (G.J.)
**\*** Correspondence: yizhe.zhang@imse.rwth-aachen.de

**Abstract:** In the face of the rapid growth in the scale and complexity of multidisciplinary systems, being able to develop reliable systems under ever-faster changing and more individual market requirements is becoming more and more challenging. The Model-Based Systems Engineering (MBSE) approach has already been researched heavily, and started to be introduced for the management of complexity, maintaining consistency, and reducing development costs and the time-to-market. However, a major drawback of the current MBSE methodologies is the lack of capability to integrate with domain-specific simulation models to investigate design concepts in the early phases of the development process. In order to address this issue, we propose a holistic system modeling approach that allows system engineers to link descriptive system models with domain-specific simulation models. In this paper, the Systems Modeling Language (SysML) is used as the standard architecture modeling language. A system modeling approach in SysML based on the system's functional architecture for system design and validation is defined. The approach was developed to integrate domain-specific models into the system model using a SysML modeler with the capability of running and reusing simulation tasks via the coupling of external tools, which helps to bridge the existing gap between models on the system level and detail level. The feasibility of the proposed approach will be evaluated based on the case study of a wind turbine (WT) system. The study shows that our approach has the potential to enable the consistent, parameter-based interlinkage of domain-specific models based on always-up-to-date data, and to assist engineers in making design decisions to meet the system requirements accurately and rapidly in different engineering fields.

**Keywords:** model-based systems engineering; wind turbine system; simulation models; seamlessness development process





## 1. Introduction

For many years, engineers have used models to represent their real-world systems for cost-effective and timely development. Normally, these models are created for a certain purpose, and thus not capturing all of the attributes of the represented system, but rather only certain specific aspects of a single domain [1,2]. As various models are used to describe the aspects of complex systems, the information of the system is dispersed in different models, and there is no information that represents the role of the model in the system. However, customer goals need to be considered within the overall design, and the system information should remain on a consistent level over all of the stages of the development. As communication between the different stakeholders plays a crucial role, an efficient communication method is needed. Nowadays, the communication in traditional approaches to system development still relies more upon document-based information exchange [3,4]. With the increasing complexity of the system, it is difficult to guarantee that the models and model input data are up-to-date and have a unified definition across disciplines.

As a modern approach that differs from the document-based approach to systems engineering, Model-Based Systems Engineering (MBSE) describes a system in models that are linked in an object-oriented, graphical, and visual system modeling language to describe the aspects of the system development process, such as the system's requirements, design, and validation in the system models [5]. MBSE aims to enhance the complexity management, specification, communication, and traceability in the product development process (PDP). Over the entire life-cycle of a product, MBSE thus focuses on supporting domain models by means of information exchange between engineers. The support during the product development is enabled using a central system model [6]. The system model can be used as an abstract digital representation of the physical, digital or cyber-physical product under development. Aiming to connect the domains via the linkage of the domain-specific models instead of deriving and sharing documents during the development process, the consistency of data flows between models can be improved significantly. Besides the management of the requirements [7,8], this also improves development [7] and validation [9–11].

Even though MBSE is emerging as an important practice in multiple industries [12], several challenges need to be addressed to realize its full potential in systems design. One of the challenges is that there is a huge leap of abstraction between domain-specific models and abstract multi-disciplinary system models [13]. The current MBSE methodologies are mainly focused on high-level architecture, as well as behavior modeling and the qualitative analysis of the system, but they lack the ability to describe the details of the design and maintain consistency among domain-specific models, which makes it difficult to ensure the accuracy of the system's design. In addition, each domain involved in the development process established its own methods and optimized tools to meet domain-specific challenges [14]. Although there are many object-oriented modeling languages that try to address this issue, the current interface techniques (e.g., Functional Mock-up Interface (FMI)) [15] are not enough to fully support the data consistency between all of the aspects in multi-domain models. The strategies to maintain the consistency between models of mechanical engineering are always tool-dependent and might lose the system data among the different tools. Moreover, a number of studies only consider the consistency checking of models and model management focusing on the software engineering domain [16].

A solution to the aforementioned challenge is a systematically built, integrative system model that provides links among the different domain-specific models and describes the system with more detail. The Object Management Group defines a model as "a selective representation of some system whose form and content are chosen based on a specific set of concerns" [17]. In this case, such a system model is linked with the domain-specific models, and it maintains data consistency by providing a central data structure and the linking parameters of the domain-specific models to the central database. This system model acts as a data source as well as data storage for the tools used in the development process. By successively building model-based system architecture upon a consistent base of parameters, the system model can perform high-precision simulation tests on the system during the system validation phase, thereby solving potential conflicts in the system development process [18].

In this paper, a Systems Modeling Language (SysML) modeler as a platform to integrate the models from the different domains is used. An integrated system model with different layers of abstraction is created, thereby adding a conceptual description of the system with increasing comprehensiveness, accuracy, and consistency. Specifically, a system modeling approach based on the classic method of functional decomposition will be applied, which is found in design methodology [19–21], to create a system model for wind turbine (WT) systems that contains significant information about the system during the development process, including the system requirements, functions, specific parameters, and so on. This paper explores the way in which to integrate the existing domain-specific models in engineering software with the system model in SysML. Finally, the proposed

approach is applied to the design and validation phases of the WT systems to demonstrate its availability and efficiency.

Our research illuminates the ways in which the strongly linked nature of the information in the system model enables comprehensive traceability analyses and decision making on solution design, which allows us to meet the system requirements throughout the redesign process more accurately and faster, even with complex systems [4]. Our contribution is an approach for the development of system models with integrated domain-specific models as an enhancement to the established MBSE approach. The proposed approach enables consistency across multiple models along with the product development (e.g., modeling and validation) process and increases the data exchange between the different domains. Furthermore, the proposed approach is demonstrated in a specific engineering use case involving the development of WT systems.

This paper is organized as follows. Section 2 considers the state of the art and the related work of systems engineering. Section 3 presents the overview of our modeling concept and approach. Next, the proposed approach is detailed through the WT use case in Section 4, which enables systems engineers to design the multidisciplinary system based on SysML. We evaluate the approach in the case studies of WT systems in Section 5. Finally, Section 6 discusses the findings, the superiority, and the remaining challenges of the proposed approach, and prospects for future research. Section 7 concludes the paper.

## 2. Background

With the increase of the complexity in the later stage of system development, there are more and more implicit dependencies between designs in the multi-disciplinary system. In order to design and optimize a cost-effective system, it is becoming more and more important to maintain the seamlessness of data between the different domains. Reference [22] worked on a co-design approach to enhance engineering creativity and achieve optimal designs. Reference [23] provided an optimization framework to solve multidisciplinary design-optimization problems. For instance, to further reduce the cost of wind energy, modern WTs increasingly require co-design methods [24] between the mechanical and control domains. Reference [25] developed an integrated toolset to design and optimize WT systems in a more integrated manner. However, the proposed method still relies on close communication between designers in various domains, which is usually a challenge for the handling of more complex systems. A system development method with closer interdisciplinary cooperation must continue to evolve. MBSE is an interdisciplinary collaborative approach to developing, testing, and optimizing systems that meet customer expectations [3]. It provides a central platform for stakeholders from a range of different disciplines that enables us to improve communication efficiency during the co-design process.

Several methods of MBSE are described in a complete survey by Estefan [26]. The Object-Oriented Systems Engineering Method (OOSEM) [27] utilizes a model-based approach to represent the system, using SysML as the modeling language. It mirrors the classical "Vee" lifecycle development model of system design, which enables the system engineer to define, specify and analyze the system among various system views throughout the development process. In the case of complex mechatronic systems across multi-domain models, the Cyber MagicGrid approach was proposed by NoMagic [28]. Cyber Magic-Grid is an extension of MagicGrid [29], which is defined with three layers of abstraction (Problem, Solution, Implementation) and four pillars (Requirements, Behavior, Structure, Parameters). The detail layer and completeness of Cyber MagicGrid allow every team member to share information about their tasks during the development process.

MBSE, in these established methods, is employed to define the requirements, system physical structure, data flows, system behaviors, and test activities in most applications [4,30]. However, with the development of technology, the physical architecture of the system will be further developed, such that it is difficult to ensure the reusability of the system model architecture. In [31], the need for function-oriented system modeling and the necessity

of defining system functions through SysML elements was observed. However, there is still no standardized top-down modeling method to ensure that the high-level designs (e.g., requirements, functions) of the system can be completely inherited in the underlying solution design of the system, so as to perform high-precision simulation analysis.

A general modeling language SysML is widely used for the centralized modeling of systems engineering problems [32]. The Object Management Group developed the SysML to create and manage models of systems using well-defined blocks, diagrams, and other visual constructs [33]. The system model, as defined with SysML, has language elements to present a complete view of the interdisciplinary system under development, including its various requirements, structures, behaviors, activities, parameters, and their interactions [6,34]. The use of SysML can manage a large number of aspects and abstract a domain-specific language to a level that permits its interaction with other system models. However, although engineers can already achieve the simple coupling between some specific models [35,36] (such as FMI and commercial integration tools like ModelCenter) and the SysML model, SysML is still difficult to integrate with complex engineering domain-specific models, such as a Multibody Simulation (MBS) for Noise, Vibration and Harshness (NVH) analysis. This can lead to errors of design and worthless modeling for real-world engineering problems. More importantly, most integration standards [37] (e.g., OSLC) are only widely used in the development of software coding, or just staying in the early stage of PDP. The system developers, having a background in mechanical engineering, feel that it is difficult to implement MBSE methods. The reason is the lack of a top-down system development approach, and traceability links not being clearly established between system functions and real-world physical components in the presented approaches.

## 3. Concept and Approach

In this section, some significant SysML elements that are used in this paper will be introduced. The details on the approach for functional system modeling linked to multi-domain models will be provided, which focuses on the system design and validation processes based on the functional architecture of the SysML model.

### 3.1. System Modeling Elements

The requirement analysis needs to be made before and during the architecture design process of the system. The complex system exhibits a large number of requirements, and therefore each requirement has a unique ID and a textual representation of the requirement. In this study, the requirements are typically classified into two types at different stages in the development process. A functional requirement describes the functions and behavior information required for the system [38]. The non-functional requirement specifies the conditions or the constraints under which the solution must remain effective. In SysML, requirements are associated with other SysML elements (e.g., the value property) using various relationships, such as the satisfy and validate relationships. The non-functional requirements can be expressed through a glossary library to enable the connections. A glossary contains the prearranged definition of the names or abbreviations that translates the text description into mathematic expressions [39]. In this paper, the non-functional requirements are satisfied by the property values contained in the solution. The functional requirements are validated through functional testing based on the corresponding solutions.

The SysML block is a modular unit that can represent an element of a system, such as a component, a function, or a testing process. The blocks collect the properties or behaviors of a system, and several relationships (e.g., Association, Generalization) are specified that enable the designer to relate the blocks to one another. The blocks with a hierarchy relationship between different system levels can be presented in the block definition diagrams (BDDs), and the internal block diagrams (IBDs) describe the internal structure of a block in terms of properties and connectors between properties [40]. Therefore, the block representation of the system provides a way for system engineers to break an entire system down into a range of interacting objects.

SysML provides constructs for the depiction of a system with more details. It enables the engineer to model mathematical constraints in a system, which can be used to identify critical performance parameters and their relationships to other parameters [40]. In this study, the "SysML constraints" elements connected with the system parameters will be used to support the simulation of the system. The so-called "SysML constraints" can not only be specified by a mathematical expression (e.g., F = m × a) but also by a group of statements in a programing language (e.g., Matlab, Python). In this study, the constraints which are specified by the Matlab function scripts serve as an interface to transfer parameters (Inputs/Outputs) between the system model and domain-specific models for the system simulations.

SysML has the ability to model continuous workflows in terms of the flow of inputs, outputs, and control using an activity diagram. It is normally used for the depiction of the behaviors of system parts or the development process with a high-level view. In this paper, the sequential SysML actions are used to provide a modeling method for workflows to verify the specific simulation results and send feedback with the continuous workflow. Actions are the primary properties of the activity diagram that describe how the activity executes and transform its inputs to outputs. There are several different actions, such as "call actions" and "read actions". Call actions allow the properties of blocks to be accessed in complex systems with hierarchies. Read actions allow the activity model to obtain properties' values from structural aspects of the system [6].

### 3.2. System Modeling Based on Functional Architecture

In order to improve the applicability of MBSE and transfer the established methods of developing engineering systems to MBSE, we aim to develop a function-oriented system modeling approach. Therefore, the method of functional decomposition found in classical design methodology is applied to the system modeling approach [41]. In our modeling approach, the system function can be encapsulated as an element of the system by SysML blocks. The system elements can be decomposed into sub-elements. Similarly, functions can be decomposed into sub-functions.

The top function element can be decomposed into sub-functions which serve as the parts of one higher-level function. A system function element, or a part of it, can be delimited by a boundary, through which physical quantities can enter and leave the function as functional flows. These flows can be energy flows, material flows, or signal flows. The function of the delimited system transforms the quantities of the incoming flows to other quantities of the outgoing flows (see (1) of Figure 1). Functions are referred to as elementary functions if the transformation of the flows they represent does not physically decompose further. This decomposition approach seeks to reach down to the elementary functions [42]. A solution will inherit the functional flows from the function it fulfills. The solution describes a general effect that fulfills a function. In addition, a solution consists of the physical effect element, geometry element, logical element, material element, and further elements like cost elements [42]. These elements are defined at the parameter level and share parameters with internal and external design or analysis models (see (2) of Figure 1).

A simulation from the specific-domain model can be integrated with the system model by the constraints block, as we mentioned in Section 3.1. We explored the method of the integration of these domain-specific development models with a system model based on a SysML modeler. The SysML modeler allows us to evaluate expressions by using a programming language such as Matlab, and then the domain-specific models are incorporated using interfaces provided by the Matlab script in the meantime. Above all, the relevant parameters are associated with each other, so that the system can be co-developed in various development models from different domain-specific tools. In this paper, the system model is also regarded as the central platform for linking and managing the information of various aspects of the system.

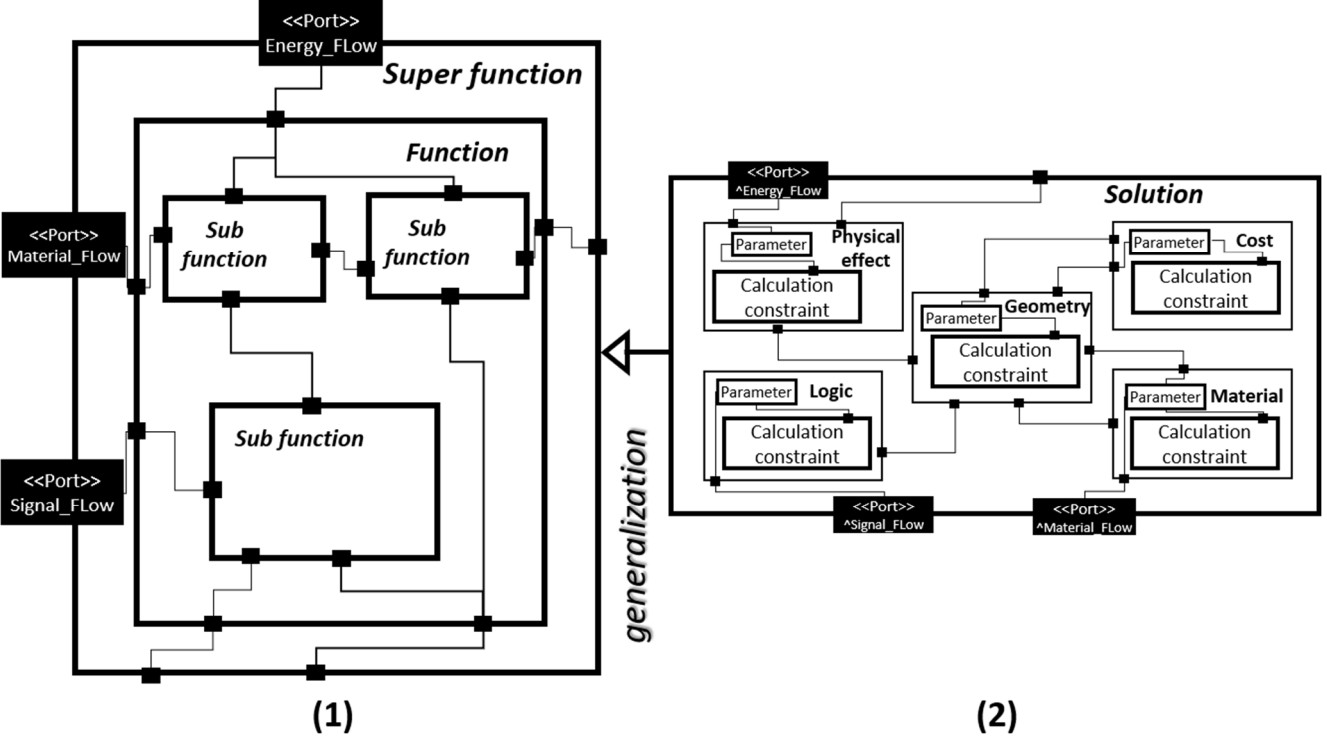

**Figure 1.** (**1**) The schematic of the function and (**2**) solution architecture in the system model.

The adopted architecture of the system model provides a sequential procedure to reduce the complexity of the system, and closes the gap between the system architecture model and domain-specific models. The functional architecture makes it possible to validate the system, and to reuse existing solutions and knowledge that support the development and technology evolution of the system. Due to the realization of functional modularization, any function and solution can be reused or replaced easily, which supports the technology evolution and increases the system development efficiency.

### 3.3. Functional Requirement Testing in the Activity Diagram

The SysML activity diagram can be used to represent the sequence of the actions (e.g., simulation actions) using control flows. In this work, the activity diagram will be used to manage the functional testing process of the system, including the simulation of domain-specific models in the related solutions, validation, and feedback (see Figure 2). In this research, the testing process is based on the functional architecture. The testing process described in actions is allowed to be sequenced according to the hierarchical architecture of system functions. A set of actions can be grouped into an activity diagram partition (also known as a swim lane) that is used to indicate responsibility for the execution of those actions [6].

As we mentioned in Section 3.2, the solutions characterize the ways in which a physical effect with a certain geometry and material properties fulfills a function. Therefore, designers need to go through a comprehensive validation of the solution at the parameter level to ensure that the selected solution satisfies the function of the system. When the testing starts, the activity diagram runs the simulations by triggering the corresponding solution with the defined system parameters. After the simulations, the significant results will be used to compare with the required values from the non-functional requirements in the validation actions. When one or more non-functional requirements are not satisfied, the corresponding solutions will be judged as not meeting the function, and the validation result of the function will be assigned as false. Finally, the validation result is saved as a property of the function, which can be used to judge whether we need to send the feedback

or continue the testing of the following functions directly. The decision node (the hollow diamond symbol) evaluates the validation results with Boolean logic. If the result is true, the outgoing flow continues the validation process to the next stage until all of the system functions are traversed. Otherwise, the feedback information about the failed functions will be sent to the stakeholders. In the next section, we will describe this method in further detail through an application in the use case of a WT system. After the functional testing process, the designer will redesign the related solutions based on the feedback information.

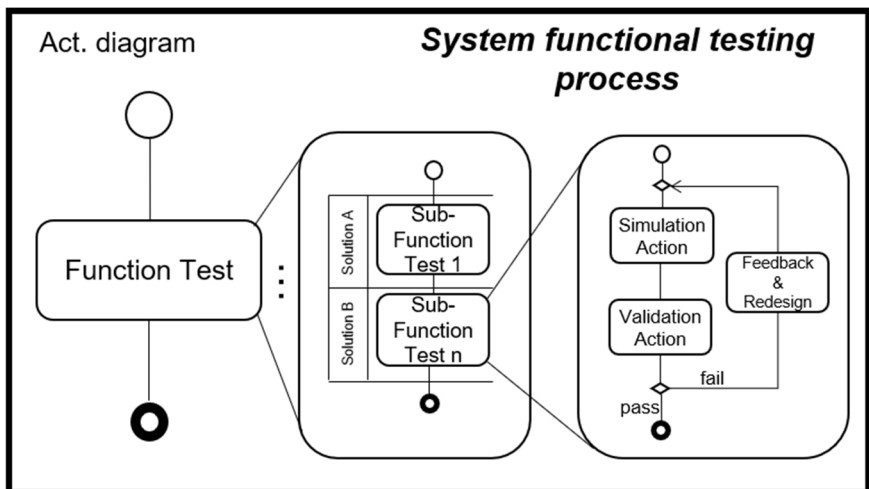

**Figure 2.** The framework of a hierarchical functional testing process in the system model.

## 4. Wind Turbine System Model Use Case

The proposed system modeling approach attempts to guide the designer in applying the variety of diagrams in SysML and having seamless modeling in the system design phase. In this paper, one of the roles of the system modeling is an approach to bring a system view that supplies the consistent model. As the realization of this approach at the system parameter level, domain-specific analysis is achieved with other kinds of tools—such as MBS software—to simulate and calculate the dynamic loads on the WT. In this paper, the proposed approach will be used to handle the complexity in the development of modern WT systems containing a large number of mechanical, control, and electrical components, as well as the corresponding models describing their behavior.

### 4.1. Wind Turbine System

A WT is a device that uses the kinetic energy of an airflow (wind energy) to produce electricity.

The main component groups of horizontal axis wind turbines (HAWT) are the rotor system, the drivetrain, and the support structure, including the tower and the control system. The rotor system is composed of the hub and blade, which converts a wind energy flow into a mechanical kinetic energy flow. The rotors of the commonly used HAWT technology rotate around the horizontal axis (see Figure 3). The drivetrain transfers the kinetic energy flow to the generator, which converts the kinetic energy flow into an electrical energy flow. In order to control the quantities of the kinetic energy flow transformed at the rotor, the pitch systems change the angle of the blades to change the aerodynamic characteristics of the rotor. The main shaft system not only transfers torque from the rotor to the rest of the drivetrain but also supports the rotor. Most WT's drivetrains include a gearbox to increase the speed of the input shaft of the generator, allowing the generator to operate in its desired operational range.

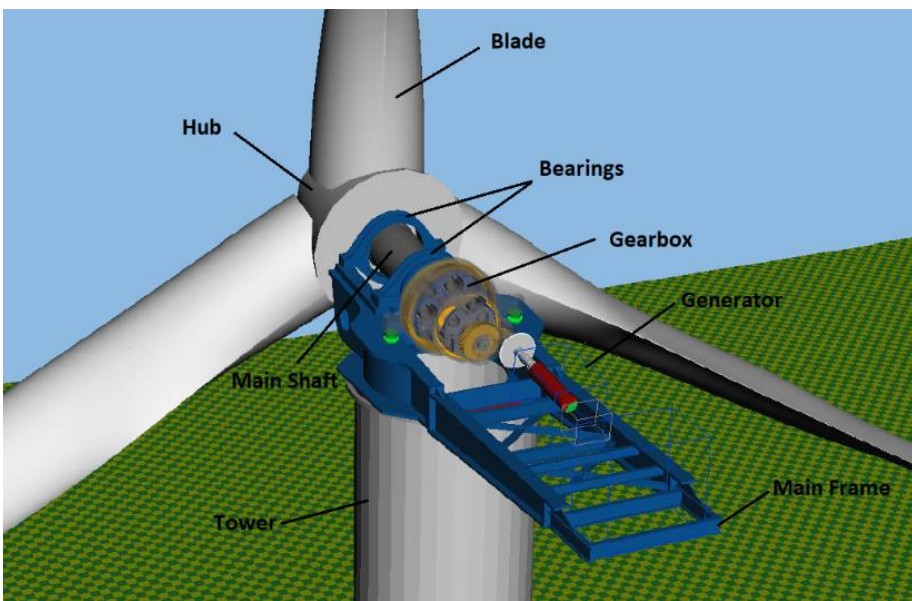

**Figure 3.** Horizontal axis wind turbine. Adapted from [43].

The control system has a major effect on the mechanical energy generated at the rotor. Depending on different wind conditions, the controller provides effective regulation to achieve optimal operation, and to ensure that the mechanical and electrical design load limits are not exceeded [44].

In the following, the models of the different domains that are considered in this use case are further described.

### 4.1.1. Mechanical Domain

Considering the large number of components and the unsteady loads, the methods of modeling and simulation are a challenge for the mechanical engineer. In order to cope with these demands, MBS is commonly utilized to conduct motion and load analysis [45]. In an MBS system, the WT system is considered as being composed of various elastic bodies that are interconnected. The MBS simulation covers the motion of those bodies, including their elasticity and their kinematic constraints, as well as the forces and moments acting in the connections between the bodies [43]. The inputs to the MBS model are the 3D geometry, the physical properties of the components, the specific contact conditions, and the characteristics of the generator and wind forces. Mechanical engineers design the system based on the required parameters and evaluate the dynamic behavior of the WT's components (e.g., deformation, speed acceleration, force, or torque). Afterward, they improve the design of the WT models based on these results.

### 4.1.2. Control Domain

Modern WTs operate at variable speeds to maximize the efficiency of the conversion, limit power during operation, maintain the grid frequency constant, and limit loads on the drivetrain. Therefore, reliable controllers are necessary to achieve a long product life [46]. Typically, the control models are developed in a proprietary programming language, such as Matlab/Simulink.

Specifically, in order to derive proper loads for the WT under different operation conditions (e.g., variable wind speed), pitch angle control is the most common solution for the adjustment of the rotation speed of the rotor and generator [47]. Therefore, the control model needs to be considered with the mechanical model, which can take place through the co-simulation of the mechanical and the control domain model [44]. The simulated variables from the mechanical model, together with the pitch angle limit, feed into the controller model, which estimates a pitch angle and feeds it back to the mechanical

model. The WT control system uses, e.g., a proportional-integral controller (PI-controller), including control parameters, the proportional coefficient ($K_P$), and the integral coefficient ($K_I$), which achieves a control loop mechanism employing feedback. In this study, the value of $K_P$ and $K_I$ is determined by the mechanical structure of the WT. In common practice, the engineers of the control domain have to check that the data is up to date manually, and vice versa. This will lead to time-wasting in the redesign process of complex systems, and even a system failure risk caused by inconsistent data.

### 4.2. Specification of the System Requirements

For the definition of the system to be modeled, first of all, the defined requirements are modeled. In Figure 4, a requirement diagram shows part of the required capabilities or conditions of the WT system under consideration, which must be satisfied. In this study, the simulation results (e.g., the cost of energy and the lifetime) are calculated and saved as value properties of the solution blocks which are connected to the non-functional requirement through the satisfy relation. The SysML tools support the visualization (e.g., highlighted in red or printing constraint failures in the console) of which non-functional requirements failed. Moreover, the system should trace its main function from the functional requirement. Therefore, for the fulfillment of the functional requirements, the system function will be checked through a function-testing activity (see Figure 4).

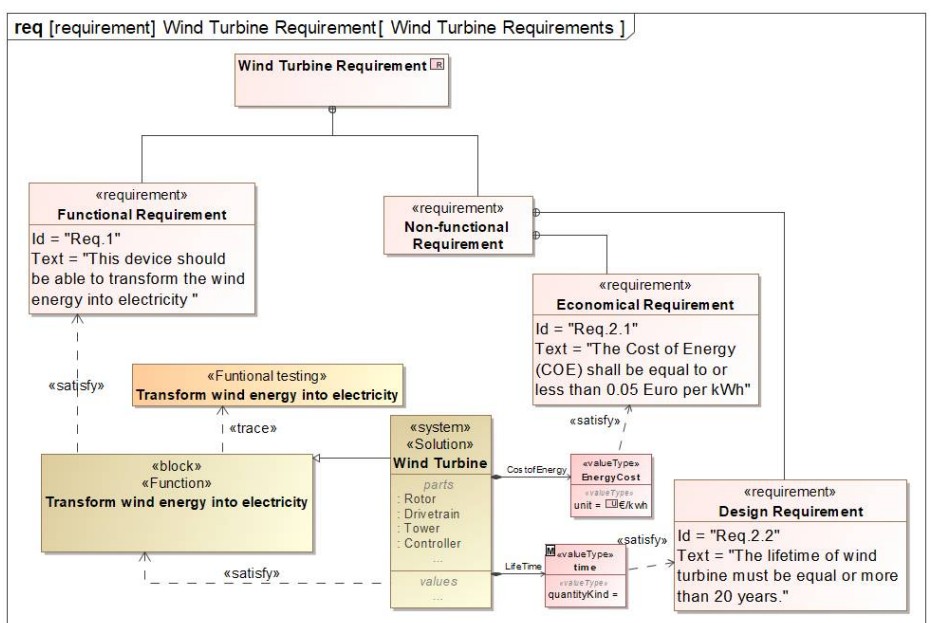

**Figure 4.** Functional and non-functional requirements of the wind turbine.

Modern requirement management tools typically maintain requirements in a database, in a table form. The SysML modeler used provides a similar mechanism to view the data using tables. It has an obvious advantage when dealing with a large number of requirements due to its compact representation. The requirement matrices can be used to represent various interrelationships between the requirements and other model elements. In this case, the matrix is applied to trace the dependency relationships of the WT, as shown in Figure 5. The relationship "Dependency" enables us to trace the related parts, meaning that if the design changes, the influence of the requirements must be considered.

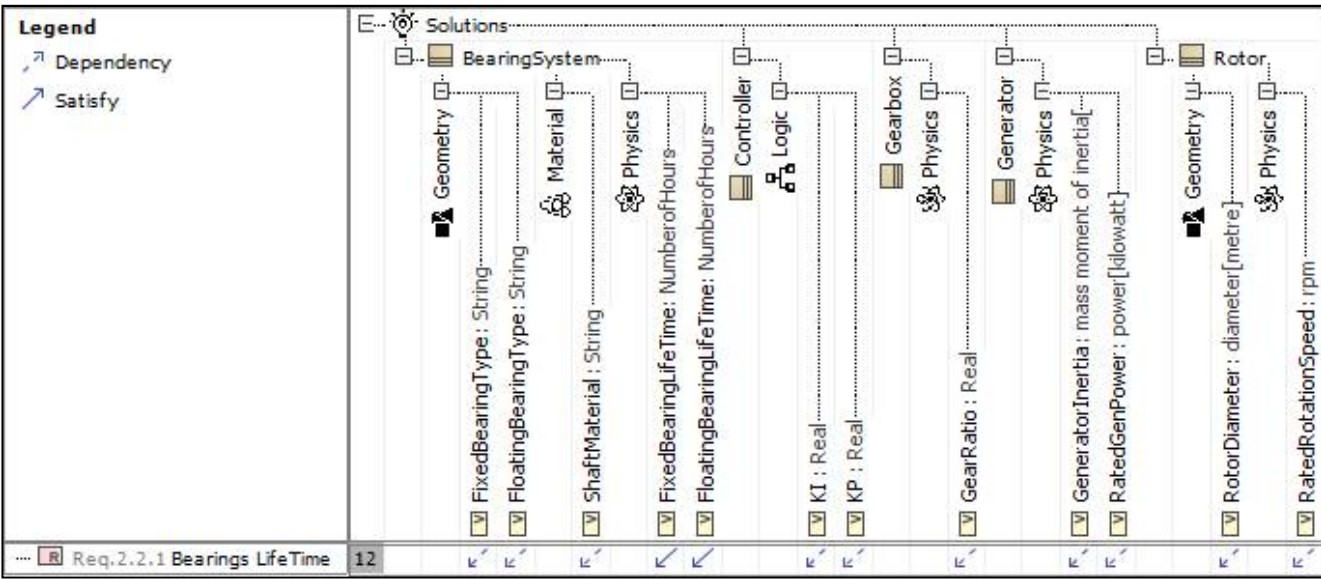

**Figure 5.** Dependency matrix of the wind turbine parameters and the bearing lifetime requirement.

It will help the designer to quickly find the solutions from a complicated system. For instance, the requirement of the lifetime of a bearing is strongly related to the component solutions of the WT in various domains (e.g., rotor, controller). The arrow icon in the matrix table will help the designer to find the key parameters quickly and show the ways towards an improvement of the economic efficiency of a WT.

### 4.3. System Architecture

In the second step, we need to be clear about the specific performances of a system; the functions can define what exactly a system does. As we mentioned in Section 4.2, the system should derive its main function from the functional system requirement, which is "Transform wind energy into electricity". In order to simplify the complexity of the WT system, the main function should be decomposed, and each sub-function should be associated with the corresponding solutions.

The functional architecture of a WT system model is illustrated in BDDs that describes the whole system by functions on different detailed levels with a hierarchical relationship. The partial functions of the WT lying on the same branch of the function architecture tree are shown on the left side of Figure 6. The right side of the figure shows the solutions corresponding to the system functions in the MBS model [43].

Moreover, the functions of the WT system are created in IBDs. IBDs describe internal functional flows between connected blocks. The connection between different components is indicated by ports in the container block, providing a common interface. Each port has a flow type, and enables us to present the flow directions. The connector between two blocks is depicted as a line connecting two symbols. This case study will focus on the energy flows between the functions, as shown in Figure 7. The energy loss flows are shown in grey, and the effective energy flows are shown in red. The function "Transform wind energy into electricity" is performed by two sub-functions. First, wind energy is transformed into mechanical energy. Then, the mechanical energy is converted into electrical energy. In the next step, the second function will be continuously broken down into several functions, such as "Transfer mechanical energy".

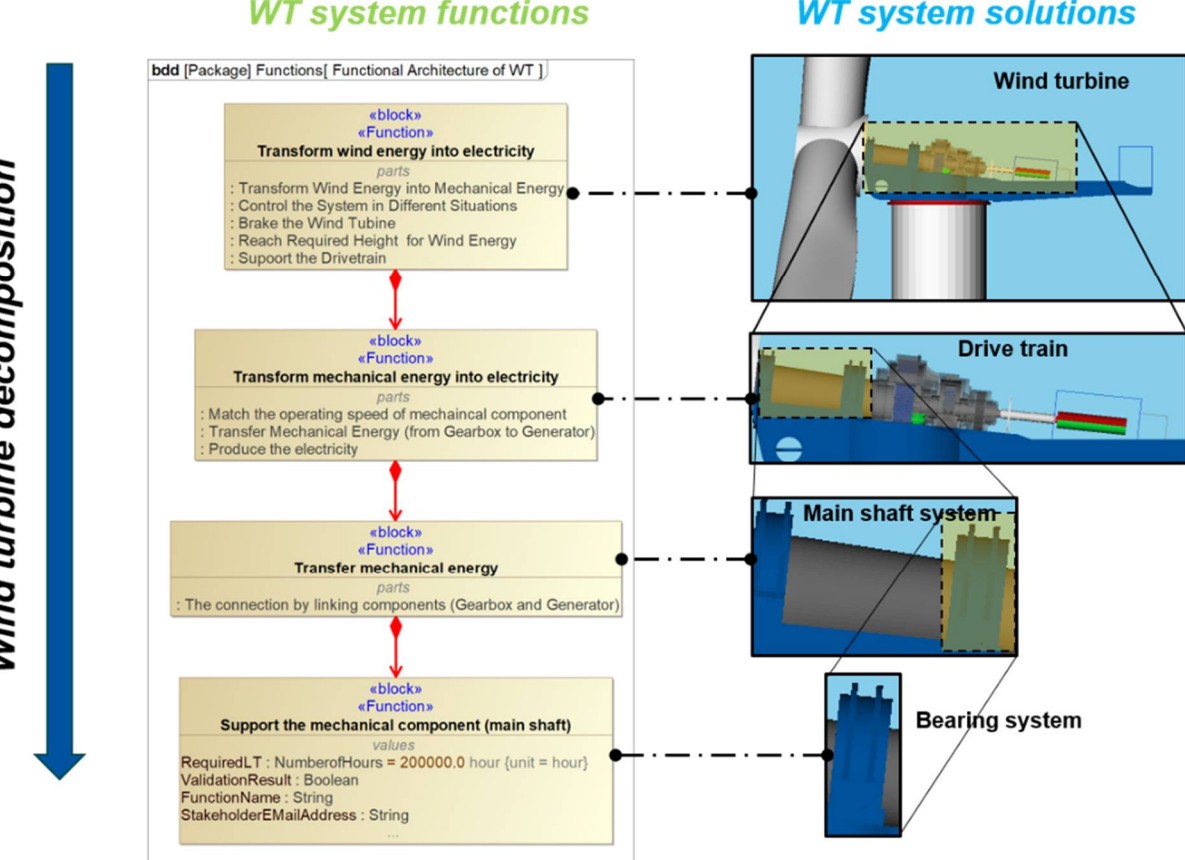

**Figure 6.** Functional decomposition of the wind turbine system and the corresponding solutions.

In addition, we created a solution for each function in IBD to support the functional testing at the parameter level. For instance, as shown in Figure 8, the solution "BearingSystem" is inherited from (generalization relationship) the sub-function "Support the mechanical component". The solution contains the bearing system's properties in terms of geometry, physical effect, material, cost domain, and simulation constraint block in the system model. This constraint delivers the design parameters in the system model to the external domain-specific design model (e.g., Lifetime Analysis Model), and then saves the calculation results back to the system model. The calculation results will be finally used for comparison with the non-functional requirements. It means that when the solution meets all of the critical non-functional requirements, we will declare that the solution satisfies the corresponding system.

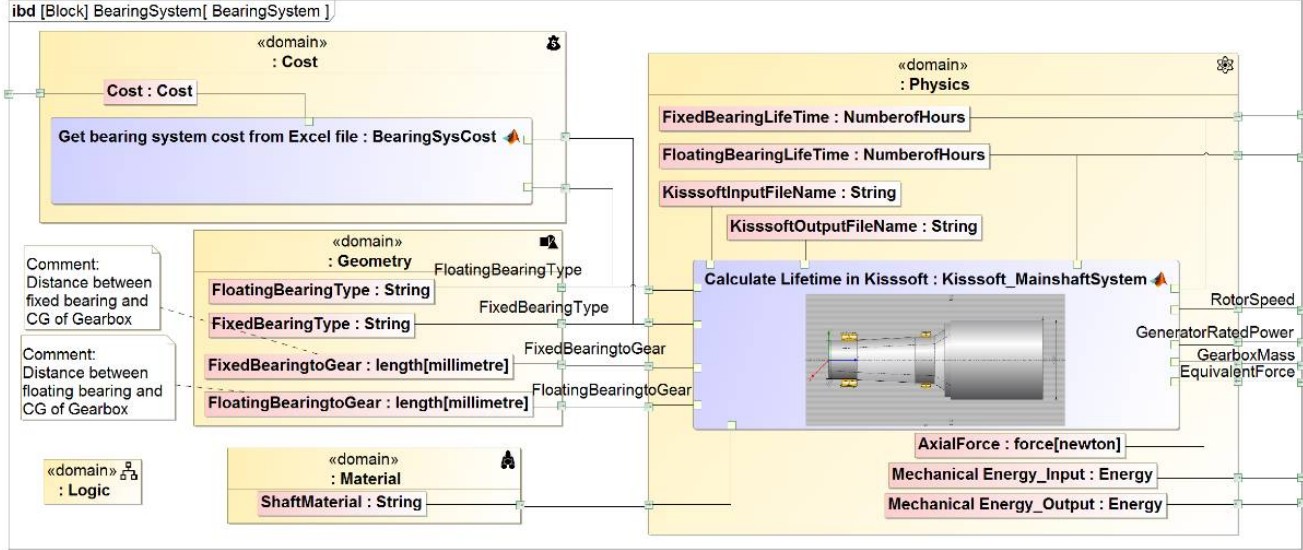

**Figure 7.** IBD of the wind turbine sub-functions' setup based on energy flows.

**Figure 8.** A sub-solution of the wind turbine system in IBDs.

Furthermore, Figure 9 shows the internal composition of the super solution "Wind Turbine System", in which the physics domain focuses on the equivalent hub load study. The constraint blocks are used to relate the various mechanical parameters from the physical effect and geometry domain with control parameters from the control domain. These parameters are distributed in the various sub-solutions of the WT system model, and are linked with the super-solution "Wind Turbine System" through the ports. In this case, the constraint block "Load calculation" calculates the hub load data, and then performs the post-processing analysis. The calculated equivalent load, as a calculated result, will be imported into SysML and stored as one of the parameters of the WT system. It can be utilized as a quantitative reference in the following validation process.

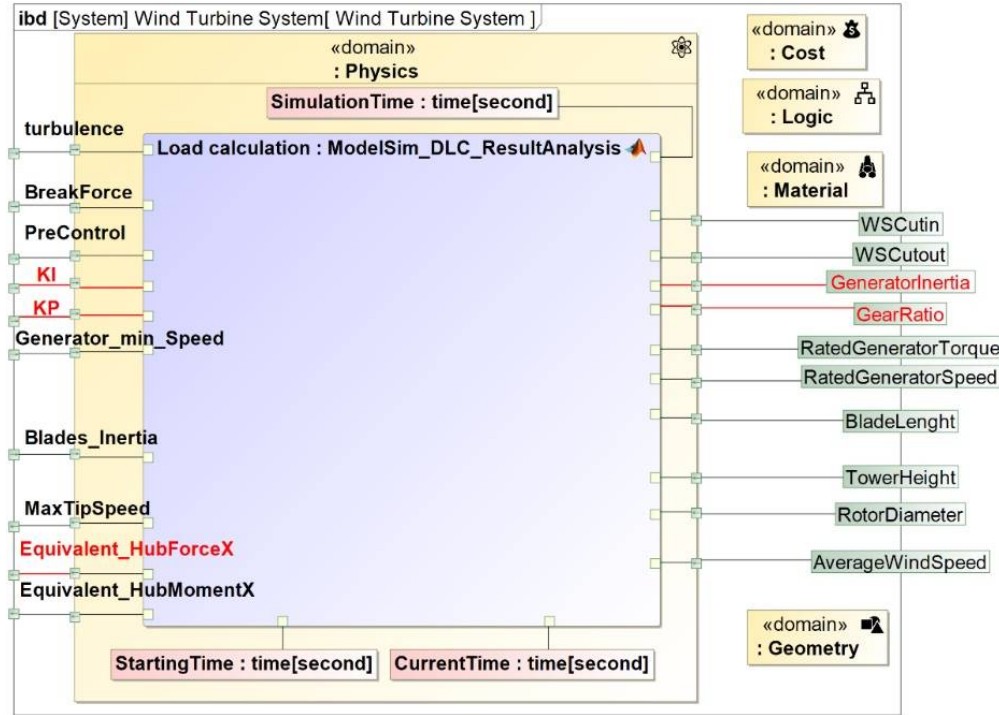

**Figure 9.** The load analysis in the wind turbine solution.

### 4.4. System Functional Testing Activity

In this study, we build tests based on activity diagrams to check the validation results of the WT system step by step. In the function testing activity, the critical simulation results from the solutions should be checked against the related design requirements, and it should then be determined whether the function is satisfied according to the validation results. For instance, the bearing lifetime plays a role as one of the criteria, which helps us to judge if the selected solution "BearingSystem" is applicable to the function based on the lifetime calculation. When the simulated lifetime is equal to or more than the required lifetime, the validation results serve as the true value, and the function "Support the mechanical component" will be regarded as passed. Figure 10 illustrates a portion of the functional testing process for a WT system. Assuming that the sub-function "Support the mechanical component" can be realized by a specific solution called "BearingSystem". When the testing starts, the activity diagram triggers the simulations of the solution "BearingSystem" based on the external simulation models. After the simulations, a significant simulated result (the lifetime value) will be used to compare with the requirement, which is "The minimum bearing rating lives for various operating and reliability conditions shall be equal or more than $2 \times 10^5$ h".

Subsequently, the decision node (the white diamond symbol) evaluates the comparison results with Boolean logic, and displays the result in the simulation console. If the lifetime value is greater than $2 \times 10^5$, then the validation result will be true, and the system function

is satisfied. The outgoing flow continues the testing process to the next function. Otherwise, it will send the feedback information to stakeholders through E-mail. According to the feedback, the solution should be replaced or the properties of the corresponding solution need to be redesigned. Finally, the functions should be tested again until the functions are passed.

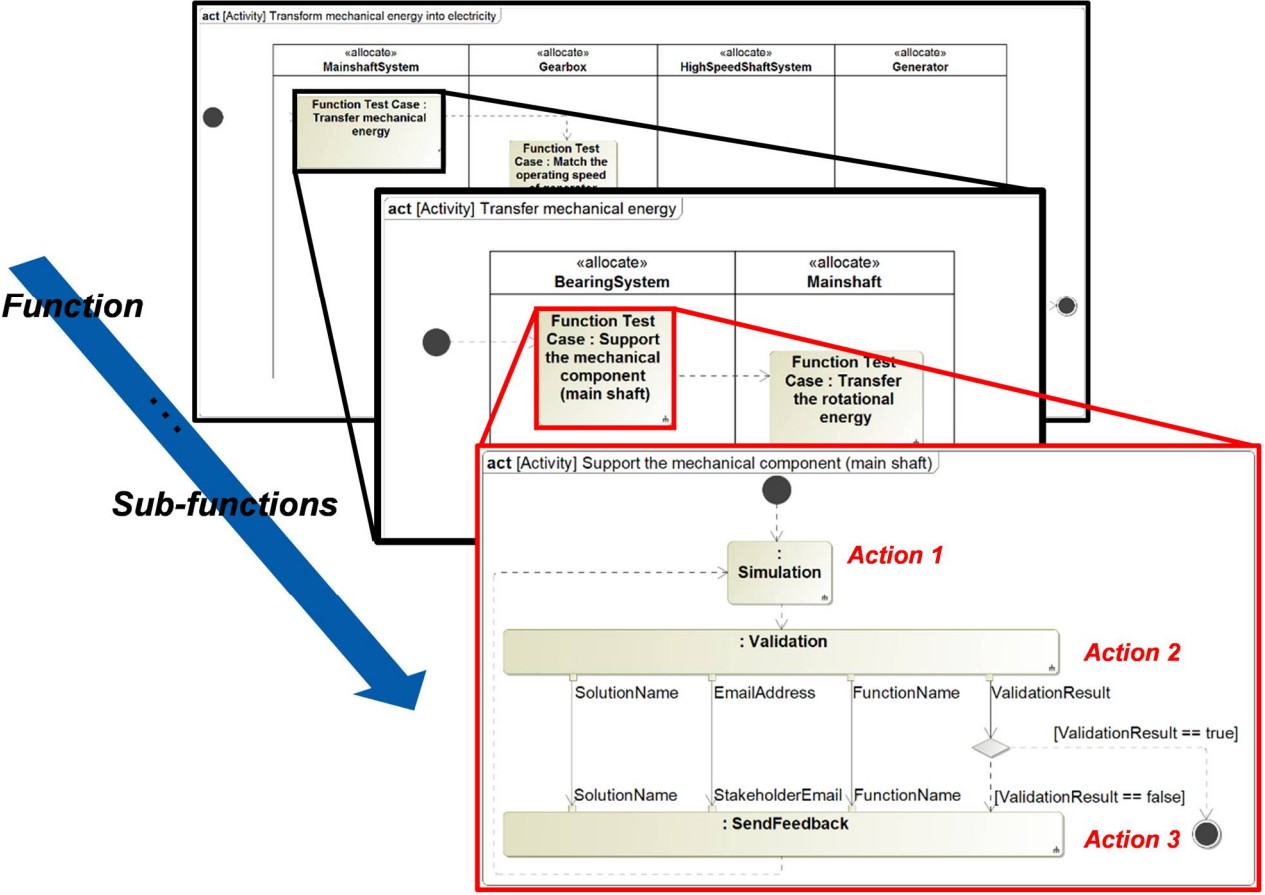

**Figure 10.** The hierarchical functional testing process of the wind turbine in the activity diagrams.

## 5. Case Studies

Our motivating example is the model-based development of the WT system. The development process case studies of the WT system in this paper are based on multidisciplinary cooperation among the involved stakeholders in the mechanical, control, and electrical domain. In the first case study, the parameters from a certain domain will be changed. The changes will lead to unsatisfactory validation results of the WT system, and will provide feedback to relevant stakeholders in time, thereby solving the design errors caused by inconsistency in the traditional development process. The second case study demonstrates that the proposed approach enables us to support an agile redesign process and make optimal decisions from the potential solutions.

### 5.1. Maintaining the Consistency of the Parameters

We assumed a redesign process of the drivetrain that increases the original generator inertia $J_{Gen\_O}$ to $J_{Gen\_O}/0.215$ and decreases the original gear ratio $i_O$ to $i_O/3$. This resulted in a new value of the drivetrain inertia ($0.92 \cdot J_{DT\_O}$) in the mechanical model. If the change is not communicated in a timely manner, members of the control domain will continue to use the obsolete control parameters $K_{I\_O}$ and $K_{P\_O}$ in their model. This leads to inconsistent parameters between the mechanical and the control domain (see Table 1).

**Table 1.** The parameters of a wind turbine in different domains under inconsistent and consistent design situations.

| | Mechanical Domain Parameters | | | Control Domain Parameters | |
|---|---|---|---|---|---|
| | Generator Inertia | Gear Ratio | Drivetrain Inertia | Integral Coefficient | Proportional Coefficient |
| Unit | $kgm^2$ | | $kgm^2$ | | |
| Original Design (Consistent) | $J_{Gen\_O}$ | $i_O$ | $J_{DT\_O}$ | $K_{I\_O}$ | $K_{P\_O}$ |
| Redesign (Inconsistent) | $\frac{J_{Gen\_O}}{0.215}$ | $\frac{i_o}{3}$ | $0.92 \cdot J_{DT\_O}$ | $K_{I\_O}$ | $K_{P\_O}$ |
| Redesign (Consistent) | $\frac{J_{Gen\_O}}{0.215}$ | $\frac{i_o}{3}$ | $0.92 \cdot J_{DT\_O}$ | $2.76 \cdot K_{I\_O}$ | $2.76 \cdot K_{P\_O}$ |

A WT system with inconsistent parameters affects the simulation results, leading to dissatisfied requirements. For instance, the above-mentioned redesign causes a significant overshoot of the rotor speed and the loads. These results are presented in Figure 11, which clearly shows that the dynamic behavior (the curve named "Inconsistent Redesign") of the WT is unstable (shown as overshoot) compared to the original design (the curve named "Consistent Original Design").

In comparison to document-based approaches, changes of these parameters will be spread to all of the stakeholders by using the proposed approach. Specifically, the mechanical engineers assigned new values to the variable "Generator Inertia" and "Gear Ratio" in the WT system model. The functional testing process triggers the co-simulation, as the simulated results are affected by these changed mechanical parameters. After that, the automatic validation of the requirements for the simulated results will be executed. In case the validation fails, a notification will be sent to each owner (e.g., the control engineer) of this view. Therefore, the control engineers can identify changes from the mechanical team and set the parameters to $2.76 \cdot K_{I\_O}$ and $2.76 \cdot K_{P\_O}$ (see Table 1) in time.

The simulation results of the redesigned WT model with consistent cooperation are illustrated in Figure 11 (the curve named "Consistency Redesign"). It clearly shows that the overshoot in the results is removed, causing a better working behavior of the WT system. Model consistency between different stakeholders ensures that the WT development process becomes more efficient.

The proposed approach enables members from the mechanical domain and members from the control domain to work together, editing all of the parameters in a common system model in SysML. Any change is transparent, enabling all of the team members to work with a consistent model. In other words, changes to the parameters of one domain can be automatically spread to other relevant domains.

*5.2. Agile Redesign Process of System Solutions*

In this case, we adopted the proposed approach to redesign the bearings of the WT system and choose the optimal solution that satisfies the corresponding function. The parameters for the system properties can be valued using an instance specification. In the first step, the system engineer can set the different instances of the solutions in an instance table containing a list of properties or using a user interface created by SysML. Instance specifications can be nested to mirror the composition of blocks [6]. Therefore, any set of values can override the initial values of the specific properties of the block, and every parameter of the WT model can be accessed and changed easily. In addition, the execution results can be recorded in a designated instance as well. Hence, it is possible that the simulation results can be reused in further validation processes. As an example, we set three instances called Bearing System A, B, and C of WT. The different types of fixed bearings and positions of assembly are determined as the variables (see Figure 12).

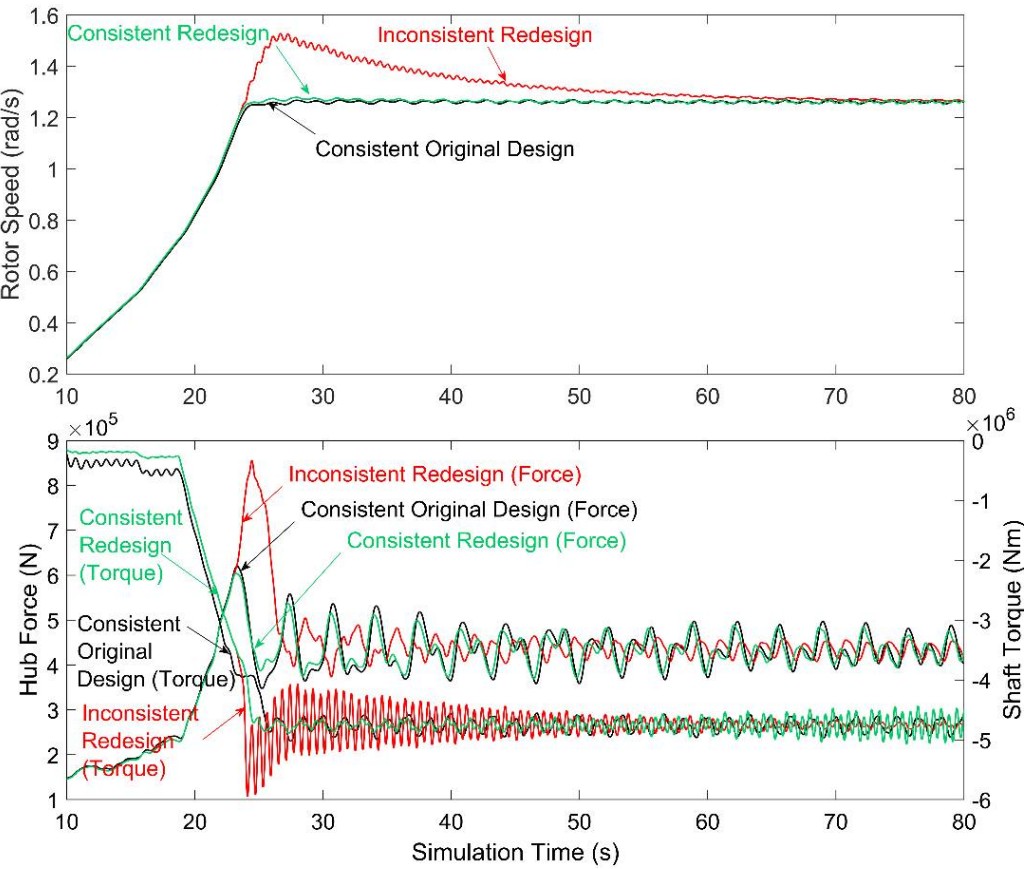

**Figure 11.** Comparison of the simulation results between those redesigned with consistent and inconsistent parameters.

| # | Name | FloatingBearingtoGear : length[millimetre] | FixedBearingtoGear : length[millimetre] | FixedBearingType : String | FloatingBearingType : String |
|---|------|---|---|---|---|
| 1 | ⊟ ⊟ BearingSystemA | | | | |
| 2 | ⊟ BearingSystemGeometryA | 3186.5 mm | 797 mm | '248/1250-B-MB' | '248/1500CAFA/W20' |
| 3 | ⊟ ⊟ BearingSystemB | | | | |
| 4 | ⊟ BearingSystemGeometryB | 3186.5 mm | 672 mm | '240/1250YMB' | '248/1500CAFA/W20' |
| 5 | ⊟ ⊟ BearingSystemC | | | | |
| 6 | ⊟ BearingSystemGeometryC | 3186.5 mm | 734.5 mm | '230/1250YMB' | '248/1500CAFA/W20' |

**Figure 12.** An instance table of bearing systems with different properties.

In order to test these solution instances, in the second step, we will execute the functional testing process, which is designed by object flows in an activity diagram. In this case, the solution "BearingSystemA" has a false value in the validation results. This means that the solutions cannot satisfy the related function, as the calculated lifetime of "BearingSystemA" is less than required. On the contrary, "BearingSystemB and C" have the true value as their result. Therefore, "BearingSystemB and C" are both feasible solutions to the corresponding function.

In the last step, we expect to find out the optimal solution from "BearingSystemB and C", because the option with "BearingSystemC" provides a lower cost than "BearingSystemB" under the same wind situation (see Figure 13). Therefore, "BearingSystemC" will be regarded as the best choice when we consider the economic aspect from the cost domain.

| # | Name | BearingSystem : BearingSystem [R] | RequiredLT : NumberofHours [V] | FixedBearingLifeTime : NumberofHours [V] | ValidationResult : Boolean [V] | Cost : Cost [V] |
|---|---|---|---|---|---|---|
| 1 | ⊞ ⊟ Support the mechanical component A | ⊟ BearingSystemA : | 200000 hour | | ☐ false | |
| 5 | ⊟ ⊟ BearingSystemA | | | | | |
| 6 | ⊟ BearingSystemPhysicsA | | | 5823.78 hour | | |
| 7 | ⊟ BearingSystemCostA | | | | | 20000 $ |
| 8 | ⊞ ⊟ Support the mechanical component B | ⊟ BearingSystemB : | 200000 hour | | ☑ true | |
| 12 | ⊟ ⊟ BearingSystemB | | | | | |
| 13 | ⊟ BearingSystemPhysicsB | | | 1000000 hour | | |
| 14 | ⊟ BearingSystemCostB | | | | | 32000 $ |
| 15 | ⊞ ⊟ Support the mechanical component C | ⊟ BearingSystemC : | 200000 hour | | ☑ true | |
| 19 | ⊟ ⊟ BearingSystemC | | | | | |
| 20 | ⊟ BearingSystemPhysicsC | | | 355558.75 hour | | |
| 21 | ⊟ BearingSystemCostC | | | | | 28000 $ |

**Figure 13.** The functional validation results with different bearing system instances.

By the proposed system modeling approach and the testing steps, our WT system achieves a seamless development process, from a sketch design based on a system model to a detail design based on domain-specific models. It shows the potential to support an efficient redesign and optimization when working with a complex system. In other words, an agile development process is possible when there is a change in the system because we can redesign and verify the system model based on the existing functional architecture with different solution instances. Moreover, the consistency feature of the system model among the view specifications ensures that the stakeholders will work in the same direction, and therefore the results in the WT development process become more accurate.

## 6. Discussion

MBSE is an approach to support the product development process for a complex system [48]. The potential of MBSE is to improve communication and knowledge reuse, leading to a reduced cycle time and lower development cost [49]. SysML provides a platform to define the high-level relationships between the requirements, functions, and physical architecture of a system. SysML v2 intends to address some of the more basic problems relating to the language, including the need for additional expressiveness, higher precision, interoperability, and improved consistency and integration [50]. However, there is still a lack of a practical system modeling approach that can easily interact with the external simulation model, instead of just focusing on the conceptual development phase or the validation phase through the low-precision constraints inside the system model. Therefore, the traditional modeling approach in SysML needs to be updated to adapt to the need of the development process in the industrial world. In addition, many domain-specific design tools need to be integrated with SysML in order to obtain consistency across the system specification [51].

The presented approach of modeling complex systems with SysML is based on the use case of WT systems. This real-world scenario is capable of representing current challenges in systems engineering, and supports the claim that MBSE has advantages over traditional document-based systems engineering [3]. Incorporating simulations allows the continuous validation of local requirements, increasing development efficiency. While the scope of the presented concept is very specific, the approach introduced is designed to be scalable, and to be extended to more general approaches and other engineering disciplines.

Another prospect of this paper is that the traceability of the advanced MBSE approach will help to quickly identify the significant parameters of the system. Achieving better complexity management results in reduced cycle times and lower costs of redesigns. The model-based design enables an efficient and flexible optimization of specific parts of the system. Therefore, the reusable parts are retained, avoiding unnecessary procedures and supporting even more agile methods by enabling the strong reuse of models. Furthermore, this paper demonstrates that SysML alone is not sufficient for sophisticated MBSE. Domain-

specific design tools and applications (e.g., MBS software) also contribute to the overall interdisciplinary system. The limitations of the integration of commercial software are clear. First of all, the commercial tools like ModelCenter are workflow builders, which are hard to integrate with all of the simulation software used in engineering fields. The multi-domain modeling languages like Modelica or Simulink will not cover all of the design fields, and cannot meet the high-precision design requirements of systems (such as lifetime and NVH analysis); therefore, the physical behavior of the WTs cannot be accurately captured. Specifically, we cannot use this type of commercial software to check the time-varying aspects of WT design, such as any phenomena related to the resonance-induced loads. Therefore, we proposed an integration approach that is based on general-purpose programming languages, such as Matlab and Python, which are designed to support a wide range of domain-specific tools. Secondly, most engineers are familiar with these languages, and there are already many existing codes. Therefore, engineers do not have to completely abandon traditional modeling methods, which are more conducive to promoting the application of the MBSE approach. The approach will be geared towards tool integration [52] to truly enable dependency tracing over the entire heterogeneous tooling landscape. The corresponding models need to be built under a unified standard, such as consistent parameter naming and formal modeling. The continuing evolution of information technology is a necessary enabler of improved modeling techniques. In the future, MBSE will benefit from the creation and reuse of model repositories, taxonomies, and design patterns [53]. The testing process adopted in this paper is based on the functional architecture of the system; therefore, the different solutions can be allocated to the same function, and can be validated at the same time. For that, the multiplicity of the parameters should be considered in the future system model to achieve the parallelization of the simulation. An effective MBSE approach requires a disciplined and well-trained team. Therefore, the efforts of the standardization of previous works and improving the understanding of MBSE for some stakeholders should be evaluated in the future.

However, the presented approach is not comprehensive. More advanced resolving mechanisms also entail additional effort for the interdisciplinary modeling teams, as dependencies have to be modeled explicitly. The solution in the system model should be further defined, and the domain-specific models should be classified in future work. The validation process should be extended with a standardized workflow to support a granular design and optimization processes. Even more specialized actions in activity diagrams provide a way to achieve this extension. For example, the simulation activities during the testing process need to be further structured. A mechanism in the simulation workflows should be established to determine the execution sequence and necessity of each simulation to save external computational costs. Other works such as data structure management and version control also need to be considered in order to obtain better and faster access to data and improve the consistent work among the stakeholders. All of the above works will have significant meanings for the further promotion of the applicability of the MBSE approach in the industrial field.

## 7. Conclusions

In this paper, an advanced Model-Based Systems Engineering (MBSE) approach using Systems Modeling Language (SysML) diagrams for system modeling and integration across domain-specific models was presented. The proposed solution covered a general concept for parameter tracing across engineering models. We applied this technique to SysML models for wind turbine (WT) systems to present the applicability of the approach to complex, modern systems and their technical parameters.

The proposed approach gives a way to apply functional decomposition concepts to system modeling; thus, system engineers will be more capable of managing system complexity, and the reusability of the model will be increased meanwhile. Additionally, the strategies to integrate domain-specific design tools from a later stage of the development

process into the solution of the system model enable system engineers to describe their systems at a higher level of abstraction while still maintaining the possibility of analyzing the system at the detailed level. This paper also presents a system of functional testing formalism using the activity diagram based on SysML. Thus, it is not only oriented to the creation of a system model but also includes engineering analysis and verification, and a validation process to support the system lifecycle.

Using this approach, the use case of a WT system was investigated to analyze the benefits of automatically distributing changes to the different stakeholders. The proposed solution reduces the engineering effort by liberating developers from manual and error-prone (mostly document-based) parameter tracing and data distribution throughout the system. This facilitates systems engineering with truly integrated models.

**Author Contributions:** Conceptualization, Y.Z., G.H., J.B., G.P. and G.J.; methodology, Y.Z., G.H, J.B. and G.J.; software, Y.Z.; validation, Y.Z.; formal analysis, Y.Z., G.H. and J.B.; writing—original draft preparation, Y.Z.; writing—review and editing, Y.Z., G.H., J.B. and G.J.; supervision, J.B. and G.J. All of the authors have read and agreed to the published version of the manuscript.

**Funding:** We thank the German Research Foundation (German: Deutsche Forschungsgemeinschaft; abbr. DFG) for funding this work as the part of the Excellence Initiative.

**Institutional Review Board Statement:** Not applicable.

**Informed Consent Statement:** Not applicable.

**Data Availability Statement:** Not applicable.

**Conflicts of Interest:** The authors declare no conflict of interest.

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
