# Peer review of "Towards Holistic System Models Including Domain-Specific Simulation Models Based on SysML"

_systems, doi:10.3390/systems9040076_

Round 1
Reviewer 1 Report
Please see the attached PDF.

Reviewer 2 Report
The paper is well structured, and the material is presented clearly. What the authors have presented is a cogent approach to model-based systems engineering. However, it is not clear to me what is unique about this approach. The idea that the system model includes key data that can be shared with other analysis tools to address different aspects of a system design is the core idea behind the US DoD digital engineering approach, and there is a growing set of standards (Open systems lifecycle management for example) and commercial tools which facilitate the type of model data exchanges described. To be a good candidate for this journal, it would be important for the paper to make clear how this material relates to these approaches, reflect the fact that this is either differs from these approaches or represents one example, and if the latter why this warrants special consideration. What is it about this implementation that makes it journal worthy?
Round 2
Reviewer 1 Report
Overall, this revision has addressed my primary concerns, and I would recommend this manuscript for publication.
Reviewer 2 Report
Revisions address earlier concerns well.